# Influence of Amino Acids on Quorum Sensing-Related Pathways in *Pseudomonas aeruginosa* PAO1: Insights from the GEM iJD1249

**DOI:** 10.3390/metabo15040236

**Published:** 2025-03-29

**Authors:** Javier Alejandro Delgado-Nungaray, Luis Joel Figueroa-Yáñez, Eire Reynaga-Delgado, Mario Alberto García-Ramírez, Karla Esperanza Aguilar-Corona, Orfil Gonzalez-Reynoso

**Affiliations:** 1Chemical Engineering Department, University Center for Exact and Engineering Sciences, University of Guadalajara, Blvd. M. García Barragán # 1451, Guadalajara 44430, Mexico; javier.dnungaray@alumnos.udg.mx; 2Industrial Biotechnology Unit, Center for Research and Assistance in Technology and Design of the State of Jalisco, A.C. (CIATEJ), Zapopan 45019, Mexico; lfigueroa@ciatej.mx; 3Pharmacobiology Department, University Center for Exact and Engineering Sciences, University of Guadalajara, Blvd. M. García Barragán # 1451, Guadalajara 44430, Mexico; eire.rdelgado@academicos.udg.mx; 4Electronics Department, University Center for Exact and Engineering Sciences, University of Guadalajara, Blvd. M. García Barragán # 1451, Guadalajara 44430, Mexico; mario.garcia@academicos.udg.mx; 5Food Engineering and Biotechnology, University Center for Exact and Engineering Sciences, University of Guadalajara, Blvd. M. García Barragán # 1451, Guadalajara 44430, Mexico; karla.aguilar5398@alumnos.udg.mx

**Keywords:** genome-scale metabolic model, quorum sensing, D- and L-amino acids, flux balance analysis, antimicrobial resistance

## Abstract

Background/objectives: Amino acids (AAs) play a critical role in diseases such as cystic fibrosis where *Pseudomonas aeruginosa* PAO1 adapts its metabolism in response to host-derived nutrients. The adaptation influences virulence and complicates antibiotic treatment mainly for the antimicrobial resistance context. D- and L-AAs have been analyzed for their impact on quorum sensing (QS), a mechanism that regulates virulence factors. This research aimed to reconstruct the genome-scale metabolic model (GEM) of *P. aeruginosa* PAO1 to investigate the metabolic roles of D- and L-AAs in QS-related pathways. Methods: The updated GEM, iJD1249, was reconstructed by using protocols to integrate data from previous models and refined with well-standardized in silico media (LB, M9, and SCFM) to improve flux balance analysis accuracy. The model was used to explore the metabolic impact of D-Met, D-Ala, D-Glu, D-Ser, L-His, L-Glu, L-Arg, and L-Ornithine (L-Orn) at 5 and 50 mM in QS-related pathways, focusing on the effects on bacterial growth and carbon flux distributions. Results: Among the tested AAs, D-Met was the only one that did not enhance the growth rate of *P. aeruginosa* PAO1, while L-Arg and L-Orn increased fluxes in the L-methionine biosynthesis pathway, influencing the *metH* gene. These findings suggest a differential metabolic role for D-and L-AAs in QS-related pathways. Conclusions: Our results shed some light on the metabolic impact of AAs on QS-related pathways and their potential role in *P. aeruginosa* virulence. Future studies should assess D-Met as a potential adjuvant in antimicrobial strategies, optimizing the concentration in combination with antibiotics to maximize its therapeutic effectiveness.

## 1. Introduction

*Pseudomonas aeruginosa* is categorized as a high-priority pathogen by the World Health Organization due to its multidrug resistance (MDR), particularly in healthcare-associated infections, and the limited potential of new antibiotic treatments, posing a global threat [1]. The pathogenesis of *P. aeruginosa* is associated with quorum sensing (QS), a bacterial system that enables cell-to-cell communication through QS signals or autoinducers (AIs) that coordinate multiple gene expressions related to several virulence factors, such as biofilm, exotoxin A, rhamnolipids, pyocyanin, and lipases [2,3]. *P. aeruginosa* comprises four types of QS systems: las, rhl, pqs, and IQS. These QS systems rely on AIs; both of them are classified as acyl-homoserine lactones (AHLs), N-(3-oxo-dodecanoyl)-L homoserine lactone (3O-C_12_-HSL) and N-butanoyl-L-homoserine lactone (C_4_-HSL), as well as 2-heptyl-3-hydroxy-4(1*H*)-quinolone (PQS), and 2-(2-hydroxyphenyl)-thiazole-4-carbaldehyde (IQS), respectively [4,5].

*P. aeruginosa* PAO1 has served as a model organism for QS research and the evaluation of novel therapeutics due to our extensive knowledge of its genome [6,7]. Recent studies have highlighted the role of amino acids (AAs) in cystic fibrosis (CF) secretions in activating virulence factors. While some AAs, such as L-Arg, enhance chemotaxis and promote infections, others (Arg, Ile, Leu, Phe, Tyr, Val, and ornithine) influence robust biofilm formation [8,9]. Nevertheless, certain AAs, including L-His, L-Arg, and L-ornithine (L-Orn), exhibit anti-QS activity [10]. Additionally, enantiomeric D-AAs have emerged as innovative biofilm inhibitors due to their ability to target multiple pathways and broad-spectrum efficacy against MDR pathogens. D-AAs, such as D-Met and D-Trp, have also been studied as adjuvants to enhance antibiotic effectiveness [11,12].

Despite the interest, there is a lack of knowledge on the action of AAs in QS-related pathways because bacterial response to a given AA in the medium can vary [13]. Reprogramming of the metabolic network in *P. aeruginosa* PAO1 ensures efficient resource allocation, enhancing bacterial survival and pathogenicity. Virulence induction is known to be accompanied by a shift in central metabolism, particularly in the tricarboxylic acid cycle, the glyoxylate pathway, and fatty acid biosynthesis [14,15].

The identification of new potential targets to combat MDR requires a deep understanding of bacterial metabolism at a system level, which facilitates its application in metabolic engineering. Computational modeling of biological systems has become indispensable in this regard [16,17]. Genome-scale metabolic models (GEMs) are network-based biochemical reactions of metabolism including their gene-protein-reaction (GPR) association. GEMs provide quantitative predictions related to bacterial growth, the production of specific metabolites, and the development of new antibiotics [18,19].

The GEM reconstruction allows the generation of a stoichiometric matrix that represents reactions and metabolites within a strain. Its structure enables the exploration of the relationship between the genotype and phenotype, where the most widely used simulation technique is the Flux Balance Analysis (FBA), which is a constraint-based analysis by the stoichiometry of the biochemical reactions optimizing the system by linear programming in the steady state, using an objective function [20,21,22]. After the first GEM of *P. aeruginosa* PAO1 iMO1056 developed by Oberhardt and colleagues [23], two additional models, iPae1146 and CCBM1146, were published [24,25]. These GEMs integrated the model iMO1056 and the metabolic information available at the time of their reconstruction. However, the most recent GEM of *P. aeruginosa* PAO1 (CCBM1146) was not based on its predecessor iPae1146 but rather on iMO1056, breaking the GEM life-cycle proposed by Seif and Palsson [26].

To accurately represent the biological complexity of *P. aeruginosa* PAO1, iJD1249 is reconstructed by integrating elements from previous metabolic models, iPae1146 and CCBM1146, while incorporating the latest biochemical data from biological databases. To explore the role of AAs in QS-related pathways, in silico analysis was conducted to test the effects of D- and L-AAs on bacterial growth rate and metabolic responses, providing new insights into their potential impact on bacterial behavior.

## 2. Materials and Methods

### 2.1. Genome-Scale Metabolic Reconstruction of P. aeruginosa PAO1

The metabolic reconstruction was developed by following the protocol by Thiele and Palsson [27], along with its protocol extension, for creating a highly curated model and the most up-to-date GEM of *P. aeruginosa* PAO1. The GEM reconstruction consists of four stages: (1) draft reconstruction, (2) manual reconstruction refinement, (3) conversion from reconstruction to a mathematical model, and (4) network evaluation [28]. For stage 1, the draft reconstruction was performed by using as template resources the GEMs iPae1146 and CCBM1146 during the GEM reconstruction [24,25]. The process involved merging both GEM datasets, followed by the removal of duplicate GPRs and the creation of a model draft consistent with the GEM maturation phase [26].

Furthermore, in this draft GEM, the most recent data were integrated about both the QS systems, AHLs and PQS, that are implicated at the same time in the production of quorum quenching (QQ) enzymes PvdQ and QuiP using the *Pseudomonas aeruginosa* BioCyc database [29] and KEGG [30].

### 2.2. Refinement and Curation of the GEM P. aeruginosa PAO1

The GEM was manually curated by using the *Pseudomonas aeruginosa* BioCyc [29] and KEGG [30] databases, comprising the most updated genome and metabolic resources at the time of GEM refinement (stage 2). In addition to the aforementioned biological databases, verification of EC numbers, enzyme names, GPR associations, metabolic systems, subsystems, and pathways was carried out using the bioinformatic resources of enzymes and metabolism data BRENDA [31], Expasy [32], ExplorEnz [33], and UniProt [34]. Additionally, the GEM was refined, focusing on QS; under this paradigm, three compartments were established: cytoplasm, periplasm, and extracellular space.

This stage also involved refining the directionality and reaction stoichiometry, particularly where no specific curation existed in the *Pseudomonas aeruginosa* BioCyc database, through the use of Rhea [35], a curated knowledge base of biochemical reactions. Additionally, metabolite naming and its abbreviations were standardized using MetaNetX [36]. Metabolites whose identifiers began with numbers were modified to comply with the requirements of the MATLAB (R2024b) algorithm (The MathWorks, Inc., Natick, MA, USA), which will be discussed in the next section.

The web-based platform MetaboAnalyst 6.0 (https://www.metaboanalyst.ca; accessed on 6 November 2024) [37] was used to assign KEGG, HMDB, PubChem, and SMILES identifiers to each compound in the metabolite list composing iJD1249. The list was manually curated. The platform was used to perform a pathway analysis method in iJD1249 through over-representation analysis (ORA) to identify the metabolic pathways that have a greater overlap with the set of metabolites than expected by chance, illustrating how those metabolites interact in the biochemical setting of *P. aeruginosa* PAO1 [38].

The scatter plot was used as the visualization method, using the hypergeometric test as the enrichment method and the relative-betweenness centrality as the topology measure to match against the reference metabolome for *P. aeruginosa* PAO1, which includes all the compounds in its pathway library (retrieved from KEGG in 2023, according to MetaboAnalyst 6.0). The pathway impact is calculated as the sum of the importance scores of the matched metabolites by the sum of the importance measures of all metabolites in each pathway [39].

### 2.3. Flux Balance Analysis for Metabolic Flux Distribution in GEM

The FBA was conducted by using an in-house algorithm developed by our research group [40]. This algorithm, designed to optimize specific metabolites, was implemented in MATLAB (R2024b) (The MathWorks, Inc., Natick, MA, USA). The biochemical reactions in GEM iJD1249 were mathematically expressed in a stoichiometric matrix considering steady-state conditions (stage 3) [40]. This matrix is presented as shown in Equation (1):(1)$$\frac{dx}{dt}=S\cdot v=0$$
where **S** is the stoichiometric matrix, with dimensions m×n, m represents the mass balance for each of the **x** metabolites, and n is the number of internal fluxes. The $v[mmol{\left(gD.W.h\right)}^{-1}]$ represents the internal flux vector and the exchange fluxes considered between the bacterium and the culture medium. The linear programming technique was applied to solve the system that maximizes the microorganism’s growth function. Equation (2) represents the growth function, and it is defined in terms of microorganism biomass:(2)$$z(x_{m})=\sum_{m=1}^{Growth}d_{m}\cdot x_{m}\to biomass$$
where ${z(x}_{m})$ represents the objective function defined as biomass, considered in biochemical reaction 251 for its composition, which includes AAs, peptidoglycan, lipopolysaccharides, and nucleotides, using the stoichiometric values from Clavijo-Butiricá et al. [25]. Additionally, $d_{m}$ represents the proportion of each metabolite, and $x_{m}$ represents the metabolite’s contribution to biomass composition. Satisfying biochemical thermodynamic restrictions, the internal fluxes are non-negative, as shown in Equation (3):*v* ≥ 0(3)

The FBA is a constraint-based framework where α and β represent the lower and upper boundaries, respectively, for the exchange fluxes $b_{j}$, as depicted in Equation (4):(4)$$\alpha \leq b_{j}\leq \beta$$

The stoichiometric matrix S will be visualized through its sparsity pattern plot using the command spy(S) in MATLAB (R2024b) (The MathWorks, Inc., Natick, MA, USA), where the nonzero values are colored and the zero values are white.

### 2.4. Defining In Silico Growth Media and GEM Validation

The validation is focused on carbon source utilization on complex nutritional inputs, following the guidelines for designing realistic in silico experiments to improve the reproducibility of GEM outcomes, as outlined by Marinos et al. [41]. The selected complex nutritional inputs were Luria–Bertani (LB) medium, minimal medium (M9), and synthetic cystic fibrosis medium (SCFM), the latter of which was developed by Oberhardt et al. [42], where the lower bounds were set to the molecular concentrations for each metabolite (stage 4) (Appendix A).

The biomass maximization values for each complex nutritional input were then compared using the specific growth rates (µ) and doubling times (t_d_) reported in experimental data from the literature [24,27].

### 2.5. Evaluation of the Metabolic Influence of Amino Acids on QS-Related Pathways

The analysis was performed to understand metabolic changes in QS-related pathways when AAs are supplied, specifically D-Met, D-Ala, D-Glu, and D-Ser, additionally, L-His, L-Glu, L-Arg, and L-Orn (stage 4). Three criteria were used to select AAs: their presence in GEM iJD1249 exchange reactions, their reported changes in CF, and their known capacity to affect *P. aeruginosa* PAO1 pathogenicity [8,11]. The analysis was constrained to 5 and 50 mM for each AA in M9 medium, as described in the previous section, and compared to fluxes in M9 medium (Appendix A). These concentrations were selected based on values commonly used in in vitro experiments [11]. The main aim was to investigate how these AAs modify the specific growth rate of *P. aeruginosa* PAO1 and alter carbon flux distributions associated with virulence factor synthesis that includes alginate, pyocyanin, pyoverdine, and rhamnolipid biosynthesis, as well as QS-related pathways. The latter were established based on their strong mediation according to the base metabolites to conform AIs. These are L-homoserine and L-methionine biosynthesis, S-adenosyl-L-methionine biosynthesis, palmitate biosynthesis, and 2-heptyl-3-hydroxy-4(1*H*)-quinolone biosynthesis.

## 3. Results

### 3.1. Reconstructed GEM of P. aeruginosa PAO1: iJD1246

The draft-reconstructed GEM was composed of 1226 genes, 1070 metabolites, 1157 biochemical reactions, 212 exchange reactions, and particularly the regulatory QS which also implicates the QQ enzymes, specifically PvdQ and QuiP. After the GEM curation of *P. aeruginosa,* PAO1 was named iJD1249, following the standard in silico naming convention [43]. The three previous GEMs of *P. aeruginosa* PAO1 are compared with the system-level composition of iJD1249 shown in Table 1.

The iJD1249 composition (see Appendix A) is depicted in Figure 1. The largest metabolic systems in iJD1249, based on the number of biochemical reactions, include transport, amino acids biosynthesis and degradation, exchange, fatty acid and lipid biosynthesis, cofactors, prosthetic groups, electron carrier biosynthesis, cell wall biosynthesis, central metabolism, culture medium, virulence factor synthesis, and purine nucleotide biosynthesis. A magnified view is depicted in Figure 1 to visualize the five metabolic subsystems of virulence factor synthesis, including QS, phenazine, pyoverdine, alginate, and rhamnolipid biosynthesis, along with the pathways that comprise each of them.

The GEM iJD1246 was structured with three compartments: cytoplasm, periplasm, and extracellular space. The compartmentalization accounted for the QS system, which involves 21 reactions, 16 genes (PA0099, PA0745, PA0996, PA0997, PA0998, PA0999, PA1000, PA1032, PA1432, PA2080, PA2081, PA2385, PA2579, PA2587, PA3476, PA3479), 15 proteins (DspI, KynA, KynB, KynU, LasI, PqsA, PqsB, PqsC, PqsD, PqsE, PqsH, PvdQ, QuiP, RhlA, RhlI), and 5 metabolites implicated in biochemical reactions in the periplasmic space (3-oxo-dodecanoate, butyrate, N-(3-oxo-dodecanoyl)-L-homoserine lactone, homoserine lactone and N-butyryl-L-homoserine lactone). These metabolites are involved in two reactions (825 and 826), where the proteins PvdQ and QuiP act as endogenous quenching agents by degrading both 3O-C_12_-HSL and C_4_-HSL.

Regarding biomolecule biosynthesis, the iJD1249 includes metabolic systems such as fatty acid and lipid biosynthesis, cell wall biosynthesis, cofactors, prosthetic groups, and electron carrier biosynthesis. The structural integrity of *P. aeruginosa* PAO1 is maintained by biosynthesis which supplies the building blocks for cellular structures, including membranes and the cell wall. The stoichiometry and energy requirements for growth and sustaining biological functions are also described by biomass and maintenance reactions. The production of metabolites and cofactors that feed into different biosynthetic and degradative pathways depends on metabolic systems, including sugar nucleotide biosynthesis, siderophore and metallophore biosynthesis, and precursor metabolites and energy generation.

The pathway analysis provided insights into the metabolic composition of iJD1249 through ORA (see Appendix A for further information). The ORA plot shown in Figure 2 displays metabolic pathways and highlights the ten most representative pathways based on the robust metabolic associations with the metabolites of iJD1249. The *Y*-axis represents the negative logarithm base 10 of the *p*-value (−log_10_*p*), calculated using the hypergeometric test. A higher log_10_*p* value indicates a stronger statistical significance. The *X*-axis states the pathway impact calculated using the betweenness centrality of metabolites detected within a given pathway. Each pathway is represented as a circle, with its position on the plot determined by these two metrics. Circle sizes correspond to the impact value: the higher the impact, the bigger the circle. The color gradient provides the pathway relevance: as the circles shift upward and to the right, indicating increasing pathway impact and statistical significance, the color transitions from gray–blue to red.

The stoichiometric matrix of iJD1249 formed on the MATLAB platform is shown in Figure 3.

### 3.2. In Silico Growth Media and GEM Validation

The nutritional inputs of M9 medium, LB medium, and SCFM were optimized to ensure computational reproducibility by simulating the most precise medium composition for each one. The culture media composition consists of individual substrates in which M9 medium contains 9 components, LB medium 53 components, and SCFM 34 components (Appendix A). The iJD1249 can simulate the bacteria growth in the three mediums (Appendix A for further information). The specific growth rates in the three mediums are shown in Table 2.

### 3.3. Metabolic Influence of Amino Acids on QS-Related Pathways

The primary QS-related metabolic pathways include palmitate biosynthesis, L-homoserine biosynthesis, L-methionine biosynthesis, S-adenosyl-L-methionine biosynthesis, and 2-heptyl-3-hydroxy-4(1*H*)-quinolone biosynthesis. These pathways support the metabolic activity necessary for the production of 3O-C_12_-HSL, C_4_-HSL, and PQS, as shown in Figure 4. The differences in AA composition between LB medium and SCFM are that LB contains Asn and Gln, while SCFM contains ornithine. Both media support an active flux in virulence factor synthesis with the most notorious flux change in the QS pathway observed for PQS showing an increase of 1.855 mmol (g D.W. h)^−1^.

The specific growth rate of *P. aeruginosa* PAO1 was evaluated when D- and L-AAs were added to M9 medium (Table 3) (carbon flux results for D- and L-AAs can be found in Appendix A). None of them repressed fluxes associated with biomass production.

D-Ala, D-Glu, and D-Ser showed an increased level of 0.022 h^−1^ in µ values when 5 mM was added. On the other hand, adding 50 mM D-Ala resulted in the highest growth rate enhancement in *P. aeruginosa* PAO1 compared to the other D-AAs. A comparison of D-AAs flux values against the M9 medium flux is depicted in Figure 5. For L-methionine biosynthesis, the most significant flux changes are related to 50 mM D-Ala, which modify L-homocysteine and pyruvate production while simultaneously increasing the fluxes for Coenzyme A and L-Met synthesis (reactions 89 and 90).

D-Ser showed slight carbon flux changes, particularly with 50 mM in palmitate biosynthesis II and 2-heptyl-3-hydroxy-4(1*H*)-quinolone biosynthesis. The most notable carbon flux change was presented with 50 mM D-Ser which exhibited a reduction in the flux of exchange pqs (17.263 mmol (g D.W. h)^−1^). The addition of D- and L-Glu resulted in exact µ with both 5 and 50 mM concentrations. This similarity extended to the reactions studied with differences observed in only 74 and 89 reactions where 5 mM D-Glu completely depleted their flux. At 50 mM, both enantiomers show a decrease in pqs flux exchange (17.972 mmol (g D.W. h)^−1^).

A comparison of L-AAs’ flux values against the M9 medium flux is depicted in Figure 6. The L-methionine biosynthesis pathway exhibited carbon flux changes in biochemical reactions when 50 mM L-Arg and L-Orn were supplied. Additionally, both L-AAs increased the reaction forming L-Met (reaction 90) that involves the *metH* gene encoding B12-dependent methionine synthase. The homoserine biosynthesis pathway showed carbon flux changes when supplied 5 mM L-His was simulated. In the palmitate biosynthesis II pathway, carbon fluxes increased with 50 mM L-Arg and L-Orn, while both AAs simultaneously decreased fluxes in reversible reactions (reaction numbers greater than 1208). A similar change was observed in the S-adenosyl-L-methionine biosynthesis pathway. For the HHQ biosynthesis pathway, 50 mM L-Arg and L-Orn led to decreased fluxes (reactions 833 and 836), while increased carbon flux to the reversible biochemical reactions.

The µ value for L-Arg (1.0223 h^−1^) was the highest among the D- and L-AAs tested in silico, followed by 50 mM L-His that obtained the second-highest µ value of 1.0041 h^−1^, and L-Orn, which ranked third with a µ of 0.7583 h^−1^.

## 4. Discussion

The iJD1249 composition, analyzed through ORA, highlights key pathways including purine and pyrimidine metabolism, amino acid metabolism (Arg, Ala, Asp, Glu, Gly, Ser, Thr, Phe, Tyr, and Trp), and vitamin B6 metabolism. These findings suggest a tightly interconnected metabolic network optimized for energy efficiency and biosynthesis that aligns with knowledge about the use of AAs by *P. aeruginosa* [44]. Furthermore, fatty acid metabolism associated with AAs contributes to membrane biosynthesis, including the formation of *N*-acetyl-muramyl pentapeptide where a representative peptide consists of Ala-Glu-Lys-Ala-Ala, emphasizing the critical role of AAs in bacterial cell wall formation [13].

Nevertheless, due to bioinformatic limitations in MetaboAnalyst, not all metabolites in iJD1249 could be matched with pathway data (see Appendix A). Despite the bioinformatic tool limitations, the metabolite data derived from iJD1249 align with the required conformation of a representative biochemical metabolism for a GEM, demonstrating its robustness in simulating essential cellular functions and the metabolic adaptability of *P. aeruginosa* PAO1.

Regarding iJD1249 validation, growth simulations were performed in different culture media. M9 medium supplemented with 22.20 mM glucose obtained a µ of 0.4016 h^−1^, with a t_d_ of 103 min, which is considerably similar than those reported in continuous culture in a similar medium composition containing 20 mM glucose, which obtained a µ of 0.31 h^−1^, and a t_d_ of 134 min [45]. Other FBA studies reported a µ value of 0.85 h^−1^, while µ values observed in vitro were 0.78 and 0.91 h^−1^ [46,47].

In the SCFM, the µ obtained was 0.4055 h^−1^, with a t_d_ of 102 min that is similar to the known range of t_d_ in vivo (114–144 min) in CF patients [48]. *P. aeruginosa* pathogenic strains exhibit a wide range of growth characteristics, as described by Ruhluel et al. [49], with a t_d_ of 74 min in SCFM and a t_d_ of 66 min for CF sputum. Notably, PQS biosynthesis upregulation in SCFM has been previously reported, suggesting that *P. aeruginosa* adapts to human airways by maintaining quinolone production [50,51].

For LB medium, the obtained µ value was 0.6943 h^−1^, with a respective t_d_ of 59.90 min. In comparison, the previous model CCBM1146 obtained a µ of 0.5458 h^−1^ and a t_d_ of 76.20 min [25]. Other reported µ values in vitro are 0.3547, 0.096, and 0.094 h^−1^, with corresponding t_d_ values of 117.25, 433.48, and 441.97 min [25,52,53]. Since the iJD1249 was reconstructed to maximize *P. aeruginosa* PAO1 growth using FBA, a faster t_d_ prediction in LB medium is expected compared to previous values. Moreover, these results closely match in vivo parameters, as the three nutritional media are well characterized with specific components, enabling realistic in silico experiments for *P. aeruginosa* PAO1.

The specific growth rate varied with AA concentration. Except for D-Met, 50 mM supplementation produced higher µ values. In general, µ values were higher when L-AAs were supplied compared to those observed with D-AAs. Among the tested D-AAs, D-Met exhibited a distinct effect, as it did not affect growth relative to the µ obtained in M9 medium. Furthermore, D-Met at 5 or 50 mM did not change particular fluxes in the central metabolism of *P. aeruginosa* PAO1, nor did it impact QS-related pathways or metabolites involved in biofilm formation. Previously, D-Met (10 mM) was reported to not exhibit properties for inhibiting biofilm formation [54].

In contrast, our findings on bacterial growth enhancement using D-Ala were consistent with those reported by Rumbo et al. [55]. D-Ala at 50 mM made significant flux changes in L-methionine biosynthesis increasing the L-Met synthesis. L-Met is a key component of S-adenosyl methionine (SAM) that is an essential precursor for AHL biosynthesis (3O-C_12_-HSL and C_4_-HSL), thereby impacting the SAM biosynthesis pathway [56]. Bearing all these results, D-Ala does not look like a promising D-AA candidate for reducing *P. aeruginosa* PAO1 virulence due to the increased bacterial growth rate.

A reduction in pqs synthesis due to D-Ser supplementation could lead to decreased virulence factor expression. While limited information is available on D-Ser implications in *P. aeruginosa* PAO1 virulence, studies have tested D-AA mixtures (D-His/D-Thr/D-Trp/D-Ser at 4 mM), which reduced the biofilm [55].

While the QS system is recognized as being beyond the scope of FBA, investigating how substrate availability influences QS precursor availability remains valuable [57]. Regarding L-AAs, L-Arg, L-Orn, and L-His exhibited the highest specific growth rates and made the most significant carbon flux increases in QS-related pathways. The high µ value for L-Arg aligns with its known role as a pleiotropic nutrient and biofilm formation enhancer. Our results imply that 50 mM L-Arg (a physiological concentration range of 50–100 mM) not only maximizes growth but also induces a flux increase in QS-related pathways. This finding further supports the link between CF-associated L-Arg decrease and the adaptive use of this AA to promote biofilm formation of *P. aeruginosa* [58,59].

## 5. Conclusions

Herein, the GEM iJD1249 was reconstructed following established protocols to curate and integrate data from previous models while expanding information related to *P. aeruginosa* PAO1. Additionally, well-standardized in silico growth media were defined to improve the accuracy of the GEM using FBA. To the best of the author’s knowledge, this is the first study analyzing the impact of AAs on QS-related pathways, unveiling flux variations in response to specific D- and L-AAs. According to our results, none of the tested AAs are promising candidates for inhibiting virulence factors by themselves. Among them, D-Met was the only AA that did not increase the growth rate of *P. aeruginosa* PAO1, whereas D-Ala significantly increased it. The enantiomeric forms of Glu exhibited the same growth rate result but led to variations in fluxes of QS-related pathways. Our findings highlight the role of AAs in CF, particularly L-Arg and L-Orn at 50 mM that significantly increased fluxes in the L-methionine biosynthesis pathway, emphasizing their influence on the *metH* gene, which encodes B12-dependent methionine synthase. Given that these AAs are implicated in CF and utilized by *P. aeruginosa* to sustain high growth rates, their metabolic effects warrant further investigation.

Future studies could assess the potential of D-Met as an adjuvant in antimicrobial strategies by optimizing its concentration in combination with antibiotics to evaluate and maximize its effectiveness. Moreover, additional research is needed to determine whether *metH* downregulation could be explored as a potential target to study its impact on pathogenicity, considering its role in L-Arg and L-Orn metabolism in CF patients. Experimental validation of these hypotheses will clarify their translational potential. Given the growing interest in QS inhibition and biofilm disruption, there is an opportunity for further research on how QS-related pathways can be modulated to reduce *P. aeruginosa* PAO1 virulence.

## Figures and Tables

**Figure 1 metabolites-15-00236-f001:**
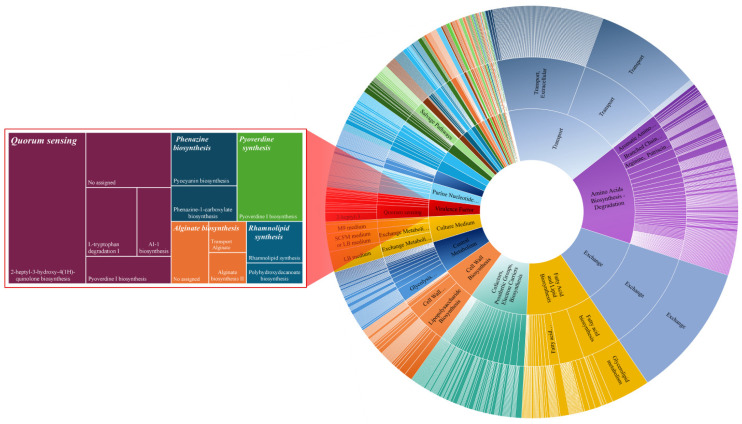
GEM composition of *P. aeruginosa* PAO1 iJD249 including metabolic systems, subsystems, and pathways focusing on virulence factor synthesis. The sunburst chart consists of three concentric circles representing hierarchical data: the innermost circle indicates the metabolic systems, the middle circle shows the metabolic subsystems, and the outermost circle represents the pathways.

**Figure 2 metabolites-15-00236-f002:**
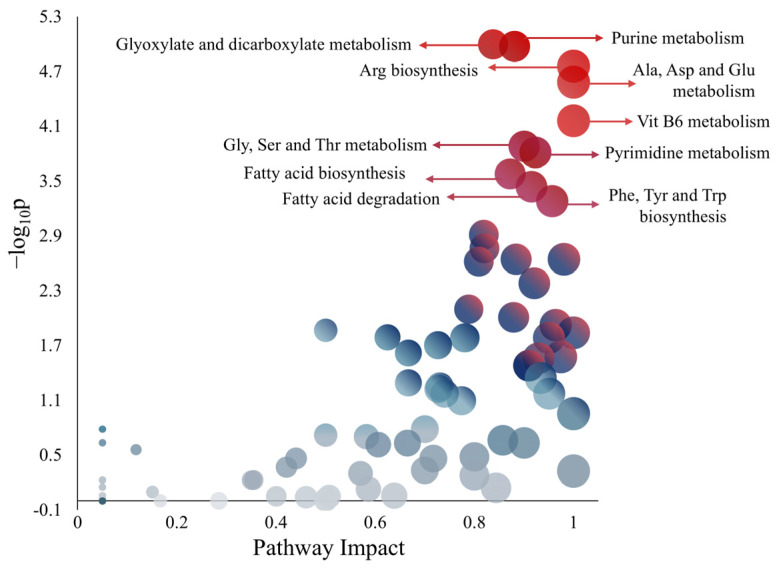
The ten most representative pathways according to the composition of iJD1249.

**Figure 3 metabolites-15-00236-f003:**
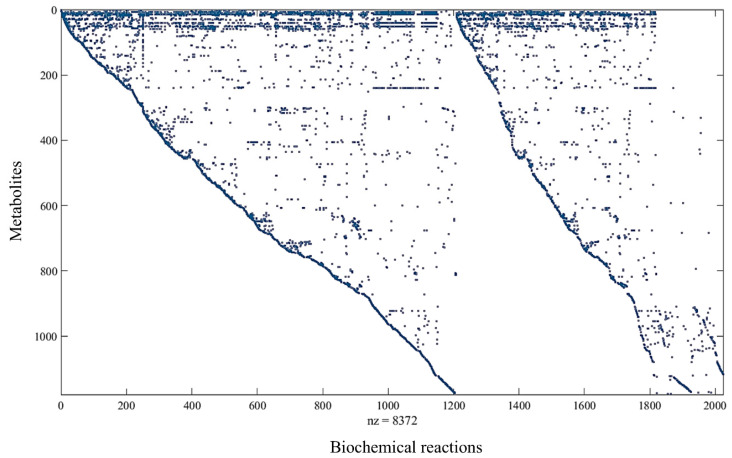
Visualization of stoichiometric matrix **S**. ‘nz’ is the number of nodes. Represents the 1178 metabolites and 1208 reactions (with 611 reversible reactions) and 205 exchange reactions.

**Figure 4 metabolites-15-00236-f004:**
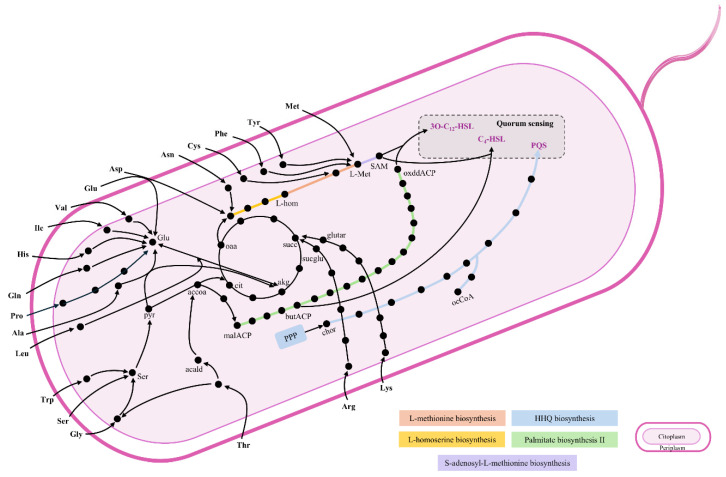
Schematic overview of primary QS-related pathways in *P. aeruginosa* PAO1, including palmitate biosynthesis, L-homoserine biosynthesis, L-methionine biosynthesis, S-adenosyl-L-methionine biosynthesis, and 2-heptyl-3-hydroxy-4(1*H*)-quinolone (HHQ) biosynthesis. The 20 AAs enable their incorporation into these pathways, supporting the production of 3O-C_12_-HSL, C_4_-HSL, and PQS. Data obtained from *P. aeruginosa* BioCyc. Abbreviations: L-hom: L-homoserine; pyr: pyruvate; malACP: malonyl-ACP; butACP: butyryl-ACP; oxddACP: 3-oxododecanoyl-ACP; sucglu: N-Succinyl-L-glutamate; chor: chorismate; ocCoA: octanoyl-CoA; SAM: S-adenosyl-L-methionine; PPP: pentose phosphate pathway.

**Figure 5 metabolites-15-00236-f005:**
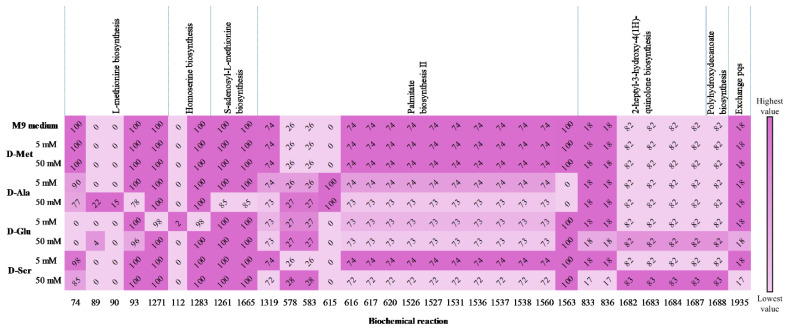
Carbon fluxes (mmol (g D.W. h)^−1^) in QS-related pathways due to 5 and 50 mM D-AA concentrations. Only reactions with flux variations are displayed. The color intensity for each biochemical reaction (columns) represents the magnitude of the flux change, with low to high values indicated in the color bar.

**Figure 6 metabolites-15-00236-f006:**
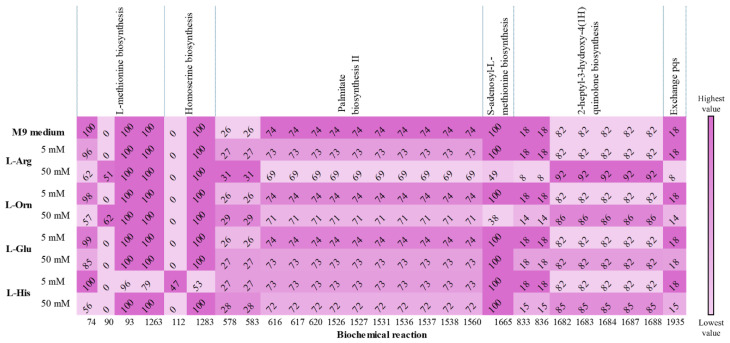
Carbon flux changes (mmol (g D.W. h)^−1^) in QS-related pathways due to 5 and 50 mM L-AA concentrations. Only reactions exhibiting flux variations are displayed. The color intensity for each biochemical reaction (columns) represents the flux change magnitude with low to high values indicated in the color bar.

**Table 1 metabolites-15-00236-t001:** GEM comparison of *P. aeruginosa* PAO1.

	Genes	Proteins	Biochemical Reactions	Exchange Reactions	Metabolites	Compartments
iJD1249	1249	1051	1208	205	1178	3 ^a^
iPae1146 [24]	1146	22	1321	172	1284	2 ^b^
CCBM1146 [25]	1146	1009	1123	120	880	2 ^b^
iMO1056 [23]	1056	1030	883	118	760	2 ^b^

^a^ cytoplasm, periplasm, and extracellular space; ^b^ cytoplasm and extracellular space.

**Table 2 metabolites-15-00236-t002:** Specific growth of *P. aeruginosa* PAO1.

Medium	µ (h^−1^)	t_d_ (h)
M9 (4 gL^−1^ glucose)	0.4016	1.7260
SCFM	0.4055	1.7094
LB	0.6943	0.9983

**Table 3 metabolites-15-00236-t003:** Specific growth rate of *P. aeruginosa* PAO1 when D- and L-Aas are supplied into M9 medium.

	µ (h^−1^)	
Amino Acid	5 mM	50 mM
D-Met	0.4016	0.4016
D-Ala	0.4236	0.6211
D-Glu	0.4236	0.6171
D-Ser	0.4236	0.6047
L-Orn	0.4455	0.7583
L-His	0.4675	1.0041
L-Glu	0.4236	0.6171
L-Arg	0.4894	1.0223

## Data Availability

The original contributions presented in the study are included in the article/Appendix A, further inquiries can be directed to the corresponding author.

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
