# Peer review of "Influence of Amino Acids on Quorum Sensing-Related Pathways in Pseudomonas aeruginosa PAO1: Insights from the GEM iJD1249"

_metabolites, 2025, doi:10.3390/metabo15040236_

Round 1
Reviewer 1 Report
Comments and Suggestions for Authors
In this manuscript, Javier et al. reconstructed the genome-scale metabolic model of P. aeruginosa PAO1 to investigate the metabolic roles of AAs in quorum sensing-related pathways. Overall, the work is interesting and provides valuable insights. Below are my comments and concerns:
- The study focused on a small selection of amino acids (D-Met, D-Ala, D-Glu, D-Ser, L-His, L-Glu, L-Arg, and L-Ornithine) at two concentrations (5 and 50 mM). This limited range might not fully capture the complexity of amino acid effects on quorum sensing-related pathways in Pseudomonas aeruginosa.
- The model reconstruction seems a manual model reconstruction that is too primitive in the era of biological databases programmatical accession. The pipeline describing the model reconstruction is too approximate and needs to be further detailed. The steps of GEM construction should be mentioned or illustrated in the figure will be easier to understand.
- Although the study provides insights into the metabolic role of AA in quorum sensing, it doesn't address the broader metabolic interactions or other potential pathways affected by these amino acids.
- More information on the molecular mechanisms behind the metabolic shifts towards quorum sensing would enhance the overall understanding.
Author Response
Manuscript ID: metabolites-3545313
Title: Influence of Amino Acids on Quorum Sensing-related Pathways in Pseudomonas aeruginosa PAO1: Insights from the GEM iJD1249
Authors: Javier Alejandro Delgado-Nungaray, Luis Joel Figueroa-Yáñez, Eire Reynaga-Delgado, Mario Alberto García-Ramírez, Karla Esperanza Aguilar-Corona, and Orfil Gonzalez-Reynoso.
Dear Assistant Editor Betty Wu and Reviewers,
We would like to express our gratitude for your professional and thorough peer review of our manuscript. We have revised our manuscript and considered all of your suggestions, which have significantly contributed to refining our work. We believe that thanks to your valuable comments, the revised version now meets the high standards of the Metabolites.
Sincerely,
The authors
Reviewers’ Comments and Authors’ Responses
In the italic text: the reviewer’s comments.
In black notation: authors’ responses
In red notation: major changes incorporated in the manuscript.
Reviewer #1: In this manuscript, Javier et al. reconstructed the genome-scale metabolic model of P. aeruginosa PAO1 to investigate the metabolic roles of AAs in quorum sensing-related pathways. Overall, the work is interesting and provides valuable insights. Below are my comments and concerns:
- The study focused on a small selection of amino acids (D-Met, D-Ala, D-Glu, D-Ser, L-His, L-Glu, L-Arg, and L-Ornithine) at two concentrations (5 and 50 mM). This limited range might not fully capture the complexity of amino acid effects on quorum sensing-related pathways in Pseudomonas aeruginosa.
We appreciate the reviewer’s understanding of the relevance of our insights into P. aeruginosa PAO1. We also thank you for pointing out the D- and L-AAs selection and the two concentrations. This information was incorporated in the subsection 2.5 Metabolic influence of amino acids on QS-related pathways (page 4, lines 170-172, and 174-175).
Three criteria were used to select AAs: their presence in GEM iJD1249 exchange reactions, their reported changes in CF, and their known capacity to affect P. aeruginosa PAO1 pathogenicity [8,11].
These concentrations were selected based on values commonly used in in vitro experiments [11].
- The model reconstruction seems a manual model reconstruction that is too primitive in the era of biological databases programmatical accession. The pipeline describing the model reconstruction is too approximate and needs to be further detailed. The steps of GEM construction should be mentioned or illustrated in the figure will be easier to understand.
We appreciate your comment, as noted the GEM reconstruction was performed manually. This approach was based on validated data from previous P. aeruginosa PAO1 GEMs, as described on page 3 lines 94-101. The current limitations of bioinformatics tools for semimanual or automated reconstruction lie in data acquisition. The tools extract GPR associations from general biological databases that are effective for creating GEMs of non-model microorganisms; however, for P. aeruginosa PAO1, a well-annotated strain, the manual reconstruction remains the most effective approach to enhancing GPR associations, ensuring greater accuracy while refining strain-specific details pertinent to the bacterium [1,2].
- C. J. Norsigian, X. Fang, Y. Seif, J. M. Monk, B. O. Palsson, Nat Protoc. 15, 1–14 (2020).
- C. Tarzi, G. Zampieri, N. Sullivan, C. Angione, Trends in Endocrinology and Metabolism. 35, 533–548 (2024).
We agree with a more detailed GEM reconstruction process. Therefore, we incorporated on page 3, lines 100-106:
The GEM reconstruction consists of four stages: (1) draft reconstruction, (2) manual reconstruction refinement, (3) conversion from reconstruction to mathematical model, and (4) network evaluation [28]. For stage 1, the draft reconstruction was done by using as template resources the GEMs iPae1146 and CCBM1146 during the GEM reconstruction [24,25]. The process involved merging both GEM datasets, followed by the removal of du-plicate GPRs and the creation of a model draft consistent with the GEM maturation phase [26].
Page 3, lines 114 and 146:
(stage 2)
(stage 3)
Page 4, lines 172 and 180:
(stage 4)
- Although the study provides insights into the metabolic role of AA in quorum sensing, it doesn't address the broader metabolic interactions or other potential pathways affected by these amino acids.
Regarding your comment, we would like to clarify that the aim of this article was to examine how the AAs tested modify the specific growth rate of P. aeruginosa PAO1 and the metabolic changes on QS-related pathways (please see page 4, lines 180-184). Therefore, prospective research could explore broader metabolic interactions (page 12, lines 387-395), although this is beyond the scope of the current article.
- More information on the molecular mechanisms behind the metabolic shifts towards quorum sensing would enhance the overall understanding.
We agree with your comment and have incorporated the request on page 2, lines 66-70:
Reprogramming of the metabolic network in P. aeruginosa PAO1 ensures efficient resource allocation, enhancing bacterial survival and pathogenicity. Virulence induction is known to be accompanied by a shift in central metabolism, particularly in the tricarboxylic acid cycle, the glyoxylate pathway, and fatty acid biosynthesis [14,15].

Reviewer 2 Report
Comments and Suggestions for Authors
I have review the following manuscript, titled as “Influence of Amino Acids on Quorum Sensing-related Pathways in Pseudomonas aeruginosa PAO1: Insights from the GEM iJD1249”
- What is the primary objective of this study, and how does it contribute to quorum sensing research in Pseudomonas aeruginosa?
- Please also mention gap-study or rationalization in introduction.
- How does this study advance the understanding of quorum sensing (QS) regulation in P. aeruginosa infections, particularly in cystic fibrosis?
- What role do D- and L-amino acids play in modulating QS pathways in P. aeruginosa?
- Please recite some recent references?
- How does the study integrate Genome-Scale Metabolic Models (GEMs) in predicting metabolic alterations related to QS?
- Why your methodology is preferable than reported one. Need valid justification.
- What are the potential clinical implications of targeting quorum sensing pathways using amino acid-based interventions?
- What modifications were made to the GEM iJD1249 model, and how does it differ from previous models?
- Please mentioned your findings in discussion
- Discussion portion should be comprehensively explained.
- How was flux balance analysis (FBA) applied to assess metabolic flux distribution?
- What criteria were used to select D- and L-amino acids, and how were their effects simulated?
- How was the metabolic network validated to ensure its accuracy in predicting bacterial responses?
- What statistical or computational approaches were employed to ensure the reliability of the findings?
- Statistical data if possible should be presented in a tabulated form.?
- How did D-Met supplementation influence the quorum sensing pathways in P. aeruginosa?
- Which amino acids exhibited the strongest impact on metabolic fluxes related to virulence and QS regulation?
- What were the observed effects of L-Arg and L-Orn on methionine biosynthesis and quorum sensing modulation?
- How did alterations in the metH gene affect QS signaling and metabolic adaptation?
- What were the major differences in bacterial growth and QS regulation between low and high amino acid concentrations?
- What are the major limitations of the study, and how can they be addressed in future research?
- What experimental validation strategies could strengthen the study’s in silico findings?
- Could the metabolic modeling framework be applied to other pathogenic bacteria beyond P. aeruginosa?
- How can these findings be leveraged to design novel quorum sensing inhibitors or antimicrobial strategies?
- What are the next steps for translating these computational insights into clinical or pharmaceutical applications?
- Results and findings should be comprehensively discussed in Conclusion.
- one more suggestion Title should be more attractive
References should be updated and English should be improved
Author Response
Manuscript ID: metabolites-3545313
Title: Influence of Amino Acids on Quorum Sensing-related Pathways in Pseudomonas aeruginosa PAO1: Insights from the GEM iJD1249
Authors: Javier Alejandro Delgado-Nungaray, Luis Joel Figueroa-Yáñez, Eire Reynaga-Delgado, Mario Alberto García-Ramírez, Karla Esperanza Aguilar-Corona, and Orfil Gonzalez-Reynoso.
Dear Assistant Editor Betty Wu and Reviewers,
We would like to express our gratitude for your professional and thorough peer review of our manuscript. We have revised our manuscript and considered all of your suggestions, which have significantly contributed to refining our work. We believe that thanks to your valuable comments, the revised version now meets the high standards of the Metabolites.
Sincerely,
The authors
Reviewer’s Comments and Authors’ Responses
In the italic text: the reviewer’s comments.
In black notation: authors’ responses
In red notation: major changes incorporated in the manuscript.
(For comments requiring cited references, these are provided at the end of the authors’ response).
Reviewer #2: I have review the following manuscript, titled as “Influence of Amino Acids on Quorum Sensing-related Pathways in Pseudomonas aeruginosa PAO1: Insights from the GEM iJD1249”
- What is the primary objective of this study, and how does it contribute to quorum sensing research in Pseudomonas aeruginosa?
The main aim of our research is to explore the role of AAs in QS-related pathways, and the effects of D- and L-AAs on bacterial growth rate and metabolic responses, providing new insights into their potential impact on bacterial behavior and how substrate influences QS precursor availability (please see page 2, lines 90-95).
2. Please also mention gap-study or rationalization in introduction.
Our research tried to fill the gap in knowledge on the action of D- and L-AAs in QS-related (please see page 2, lines 65 and 66) and also palliate the breaking in the GEM life-cycle of P. aeruginosa PAO1 (please see page 2, lines 86-89).
3. How does this study advance the understanding of quorum sensing (QS) regulation in P. aeruginosa infections, particularly in cystic fibrosis?
The findings about how P. aeruginosa are particularly more virulent in CF, related to the AAs present, principally L-Arg and L-Ornithine, were that L-Arg not only maximizes growth but also induces the greatest flux increase in QS-related pathways. Also, both L-AAs increased fluxes in the L-methionine biosynthesis pathway (please see page 11, lines 369-375). We value your comment and have clarified this by restructuring lines 386-388, page 12:
Our findings highlight the role of AAs in CF, particularly L-Arg and L-Orn at 50 mM that significantly increased fluxes in the L-methionine biosynthesis pathway, emphasizing their influence on the metH gene, which encodes B12-dependent methionine synthase.
4. What role do D- and L-amino acids play in modulating QS pathways in P. aeruginosa?
The role of the tested D- and L-AAs varied, with D-Ala at 50 mM significantly altered fluxes in L-methionine biosynthesis. Additionally, D-Ser supplementation reduced pqs synthesis. Among L-AAs, 50 mM L-Arg and L-Orn not only enhanced bacterial growth but also induced the highest flux increases in QS-related pathways (please see page 11, lines 352-373 for details).
5. Please recite some recent references?
The cited references were used based on their relevance to the research and availability within the state-of-art in GEM reconstruction and the impact of D- and L- AAs on P. aeruginosa PAO1. The protocols cited in this research represent the cornerstone of the field; therefore, some references predate 2020.
6. How does the study integrate Genome-Scale Metabolic Models (GEMs) in predicting metabolic alterations related to QS?
We appreciate your comment and would like to clarify that the GEM allowed the exploration of changes in QS-related pathways. Its structure of reactions and metabolites enables simulations using the Flux Balance Analysis, which optimizes the system through linear programming by using an objective function. This approach is part of the stage of network evaluation (please see page 2, lines 78-83, and further elaborated in subsection 2.5 on page 4, lines 176-190).
7. Why your methodology is preferable than reported one. Need valid justification.
In silico experimentation relies on methodologies to generate results that closely approximate reliable in vitro experiments. The current cornerstone of the field is the establishment of standardized protocols to ensure consistency in GEM reconstruction and FBA-based simulations. The most reliable protocols were developed by Thiele and Palsson [1], along with its protocol extension, and the path to improve the life-cycle and quality of GEMs [2,3]. Additionally, defining in silico growth media for GEM validation [4].
References
- I. Thiele, B. Palsson, Nat Protoc. 5, 93–121 (2010).
- C. J. Norsigian, X. Fang, Y. Seif, J. M. Monk, B. O. Palsson, Nat Protoc. 15, 1–14 (2020).
- Y. Seif, B. Ø. Palsson, Cell Syst. 12, 842–859 (2021).
- G. Marinos, C. Kaleta, S. Waschina, PLoS One. 15 (2020), doi:10.1371/journal.pone.0236890.
8. What are the potential clinical implications of targeting quorum sensing pathways using amino acid-based interventions?
According to our results, none of the tested AAs appear to be promising candidates for directly inhibiting virulence factors. However, in the Conclusions section, we highlight the potential use of D-Met as an adjuvant in combination with antibiotics, as it did not increase the specific growth rate (please see page 11, lines 354-359). Additionally, the observed effects of L-Arg and L-Orn on growth rate and their involvement in QS-related pathways further investigation could clarify their translational, specifically, targeting genes or proteins involved in L-methionine biosynthesis pathway may provide insights into strategies to decrease P. aeruginosa PAO1 pathogenicity (see page 11, 399-402).
9. What modifications were made to the GEM iJD1249 model, and how does it differ from previous models?
The GEM iJD1249 stands out as the most up-to-date GEM of P. aeruginosa PAO1. Its composition is relevant due to its complexity, which is reflected in the number of genes, proteins, biochemical and exchange reactions, metabolites, and compartments compared to previous models (see Table 1, page 5).
10. Please mentioned your findings in discussion
We appreciate your comment. We have incorporated additional findings on page 11, lines 352–354, while the rest of our findings are further detailed in the Discussion section.
The specific growth rate varied with AA concentration. Except for D-Met, 50 mM supplementation produced higher µ values. In general, µ values were higher when L-AAs were supplied compared to those observed with D-AAs.
11. Discussion portion should be comprehensively explained.
We appreciate your comment. We have improved the clarity of the Discussion section by refining the text and adding lines 354–356 on page 11.
The specific growth rate varied with AA concentration. Except for D-Met, 50 mM supplementation produced higher µ values. In general, µ values were higher when L-AAs were supplied compared to those observed with D-AAs.
12. How was flux balance analysis (FBA) applied to assess metabolic flux distribution?
We sincerely appreciate your comment. We have restructured lines 142-146 (page 4) to clarify that FBA was applied in MATLAB using GEM iJD1249 to obtain the stoichiometric matrix based on its composition.
The FBA analysis was conducted by using an in-house algorithm developed by our research group [40]. This algorithm, designed to optimize specific metabolites, was implemented in MATLAB (R2024b) (The MathWorks, Inc., Natick, MA, USA). The biochemical reactions in GEM iJD1249 were mathematically expressed in a stoichiometric matrix considering steady-state conditions (stage 3) [40].
13. What criteria were used to select D- and L-amino acids, and how were their effects simulated?
We are grateful for pointing out the D- and L-AAs selection. This information was incorporated in subsection 2.5 Metabolic influence of amino acids on QS-related pathways (page 4, lines 170-172).
Three criteria were used to select AAs: their presence in GEM iJD1249 exchange reactions, their reported changes in CF, and their known capacity to affect P. aeruginosa PAO1 pathogenicity [8,11].
The effects of D- and L-AAs were simulated by individually supplying them into M9 medium and performing FBA in MATLAB (see page 4, lines 182-184).
14. How was the metabolic network validated to ensure its accuracy in predicting bacterial responses?
The validation was based on carbon source utilization, specifically Luria-Bertani medium, minimal medium, and synthetic cystic fibrosis medium. The specific growth rates obtained from these media were compared with µ values and doubling times reported in experimental studies to ensure result accuracy (see page 4, lines 166-176).
15. What statistical or computational approaches were employed to ensure the reliability of the findings?
We thank you for your comment. The reliability of our findings is based on the deterministic nature of FBA and linear programming, which solve the objective function and find its optimal solution [1,2]. Validation was conducted through carbon source utilization, specifically Luria-Bertani medium, minimal medium, and synthetic cystic fibrosis medium. The specific growth rates obtained from these media were compared with µ values and doubling times reported in experimental studies to ensure result accuracy (see page 4, lines 166-176).
References:
- M. MacGillivray et al., Sci Rep. 7 (2017), doi:10.1038/s41598-017-00170-3.
- J. D. Orth, I. Thiele, B. O. Palsson, Nat Biotechnol. 28, 245–248 (2010).
16. Statistical data if possible should be presented in a tabulated form.?
We sincerely appreciate your comment. The results from simulations using LB, M9, and SCFM media are presented in Table 2 (see page 8, line 263). Additionally, results for the tested AAs are provided in Table 3 (page 9, line 287).
17. How did D-Met supplementation influence the quorum sensing pathways in P. aeruginosa?
We appreciate your question. Carbon flux changes in QS-related pathways are illustrated in Figure 5 (page 9, line 295) and further discussed on page 11, lines 358-360: “Furthermore, D-Met at 5 or 50 mM did not change particular fluxes in the central metabolism of P. aeruginosa PAO1, nor did it impact QS-related pathways or metabolites involved in biofilm formation.”
18. Which amino acids exhibited the strongest impact on metabolic fluxes related to virulence and QS regulation?
We appreciate your inquiry. To clarify this point, we have incorporated the relevant information on page 11, lines 385–386:
L-Arg, L-Orn, and L-His exhibited the highest specific growth rates and made the most significant carbon flux increases in QS-related pathways.
19. What were the observed effects of L-Arg and L-Orn on methionine biosynthesis and quorum sensing modulation?
We appreciate your inquiry regarding methionine biosynthesis and other QS-related pathways. To clarify this point, we have incorporated the relevant information on page 10, lines 308–319.
The L-methionine biosynthesis pathway exhibited carbon flux changes in biochemical reactions when 50 mM L-Arg and L-Orn were supplied. Additionally, both L-AAs increased reaction forming L-Met (reaction 90) that involves the metH gene encoding B12-dependent methionine synthase. The homoserine biosynthesis pathway showed carbon flux changes when supplied 5 mM L-His was simulated. In the palmitate biosynthesis II pathway, carbon fluxes increased with 50 mM L-Arg and L-Orn, while both AAs simultaneously decreased fluxes in reversible reactions (reaction numbers greater than 1208). A similar change was observed in the S-adenosyl-L-methionine biosynthesis pathway. For the HHQ biosynthesis pathway, 50 mM L-Arg and L-Orn led to decreased fluxes (reactions 833 and 836), while increased carbon flux to the reversible biochemical reactions.
20. How did alterations in the metH gene affect QS signaling and metabolic adaptation?
Carbon fluxes changed when 50 mM L-Arg and L-Orn were supplied, with the most notable shift occurring in the L-methionine biosynthesis pathway in which the reaction associated with the metH gene increased, enhancing carbon flux distribution in QS-related pathways. This information has been incorporated on page 12, lines 404-407:
Our findings highlight the role of AAs in CF, particularly L-Arg and L-Orn at 50 mM that significantly increased fluxes in the L-methionine biosynthesis pathway, emphasizing their influence on the metH gene, which encodes B12-dependent methionine synthase.
21. What were the major differences in bacterial growth and QS regulation between low and high amino acid concentrations?
We appreciate your question, the role of the tested D- and L-AAs varied, as answered in the reviewer’s comment number 4. We have incorporated the differences observed between low and high AA concentrations (5 and 50 mM) to clarify this (page 11, lines 352-354).
The specific growth rate varied with AA concentration. Except for D-Met, 50 mM supplementation produced higher µ values. In general, µ values were higher when L-AAs were supplied compared to those observed with D-AAs.
22. What are the major limitations of the study, and how can they be addressed in future research?
The major limitation of this research was the complexity of data analysis due to the large dataset generated from each AA and its two concentrations. Developing advanced data visualization would improve the representation and interpretation of such extensive findings. To address this limitation, we incorporated data visualization figures to depict carbon flux changes, as shown in Figures 5 and 6.
23. What experimental validation strategies could strengthen the study’s in silico findings?
Validation strategies for FBA are limited, with carbon source utilization strategy (comparison of growth rates on different substrates) being one of the most commonly applied approaches. This strategy was implemented in our research, and it has been applied in several studies, showing agreement with experimental data. Further validation strategies, such as 13C-Metabolic Flux Analysis or experimental testing (wet lab) of our generated hypotheses, could be employed to assess the reliability of our in silico findings [1].
References:
- J. A. M. Kaste, Y. Shachar-Hill, Biotechnol Prog. 40 (2024), doi:10.1002/btpr.3413.
24. Could the metabolic modeling framework be applied to other pathogenic bacteria beyond P. aeruginosa?
The iJD1249 model can be used as a GPR association template for other pathogenic bacteria as an initial approach to generate new GEMs; however, a key limitation is the biological capacity of the specific strain to accurately represent the biological system under study [1,2]. This approach offers potential but requires further refinement and curation, and may be particularly helpful for strains of Pseudomonas aeruginosa.
References:
- C. J. Norsigian, X. Fang, Y. Seif, J. M. Monk, B. O. Palsson, Nat Protoc. 15, 1–14 (2020).
- C. Tarzi, G. Zampieri, N. Sullivan, C. Angione, Trends in Endocrinology and Metabolism. 35, 533–548 (2024).
25. How can these findings be leveraged to design novel quorum sensing inhibitors or antimicrobial strategies?
We sincerely appreciate your comment. We identified shifts in QS-related pathways for AAs tested. A new approach should focus on decreasing the flux within these pathways by identifying and targeting molecules that interfere with the involved genes or proteins (see page 12, lines 399-407).
26. What are the next steps for translating these computational insights into clinical or pharmaceutical applications?
Our findings highlight two approaches. First, D-met does not increase the growth rate of the bacterium. This shed some light on why this D-AA has been used as an adjuvant with antibiotics. This result suggests a path forward for optimizing its concentration with antibiotics to enhance therapeutic responses. Second, the metH gene was observed to change carbon flux when L-Arg and L-Orn were tested at 50 mM, both of which are associated with CF. This opens a new avenue for investigating how modulating this gene could impact the pathogenicity of P. aeruginosa.
27. Results and findings should be comprehensively discussed in Conclusion.
The conclusion section has been refined based on the valuable comments and insightful queries from the reviewers. We hope that these revisions have made the section more comprehensive and clearer for readers.
28. one more suggestion Title should be more attractive
We sincerely appreciate your suggestion regarding the title. However, we believe that the current title accurately reflects the scope and focus of our research. It effectively conveys the study’s key elements, including the influence of amino acids on QS-related pathways and the development of the GEM iJD1249 for P. aeruginosa PAO1.
